# Hexagonal-Shaped Spin Crossover Nanoparticles Studied by Ising-Like Model Solved by Local Mean Field Approximation

**Catherine Cazelles** [1], **Jorge Linares** [2,3,*], **Mamadou Ndiaye** [2], **Pierre-Richard Dahoo** [4]
**and Kamel Boukheddaden** [2,*]

1   Université Paris-Saclay, UVSQ, IUT de Mantes en Yvelines, 78200 Mantes la Jolie, France; catherine.cazelles@uvsq.fr
2   Université Paris-Saclay, UVSQ, CNRS, GEMAC, 78000 Versailles, France; mamadou.ndiaye@uvsq.fr
3   Departamento de Ciencias, Sección Física, Pontificia Universidad Católica del Perú, Apartado 1761, Peru
4   Université Paris-Saclay, UVSQ, CNRS, LATMOS, 78290 Guyancourt, France; pierre.dahoo@uvsq.fr
*   Correspondence: jorge.linares@uvsq.fr (J.L.); kamel.Boukheddaden@uvsq.fr (K.B.)

**Abstract:** The properties of spin crossover (SCO) nanoparticles were studied for five 2D hexagonal lattice structures of increasing sizes embedded in a matrix, thus affecting the thermal properties of the SCO region. These effects were modeled using the Ising-like model in the framework of local mean field approximation (LMFA). The systematic combined effect of the different types of couplings, consisting of (i) bulk short- and long-range interactions and (ii) edge and corner interactions at the surface mediated by the matrix environment, were investigated by using parameter values typical of SCO complexes. Gradual two and three hysteretic transition curves from the LS to HS states were obtained. The results were interpreted in terms of the competition between the structure-dependent order and disorder temperatures ($T_{O.D.}$) of internal coupling origin and the ligand field-dependent equilibrium temperatures ($T_{eq}$) of external origin.

**Keywords:** nanomaterials; spin-crossover; Ising-like model; local mean-field approximation; matrix effects

## 1. Introduction

Over the last few decades, the storage capacity and downsizing of electronic components have been of concern for potential industrial applications in accordance with Moore's law. The search for new components scaled to dimensions at the nanometer level and materials more efficient for data storage is a major economic issue that has driven research towards alternatives to current silicon technologies. This quest was behind the proposal that emerged to use a molecule as an active element in an electronic device. Molecular electronics, based on the taming and manipulation of electrical, optical, or magnetic signals in devices composed of molecules, is continuously being developed at an ever-increasing rate [1]. In this context, spin transition materials have shown great potential and aroused much interest [2–5]. Indeed, these compounds have two different stable magnetic states termed high spin (HS) and low spin (LS). They can switch reversibly and while under control from one spin state to another. This transition is generally caused by a change in temperature [6–9]. It has been shown that it is also sensitive to a wide spectrum of external stimuli such as pressure [3,10–12], visible light [13–18], and magnetic field [19].

The change in spin state is accompanied by significant changes in the color, volume, magnetic state, and electrical conductivity of the compound, which can be observed by using different characterization techniques (such as magnetometry, Mössbauer, optical spectroscopy, X-ray diffraction, and calorimetry) [20–24].

The phenomenon of molecular bistability can occur for a certain number of metal ions belonging to the first series of transition metals and, more particularly, for metals with the $3d^4$–$3d^8$ configurations. For example, Fe (II) [25,26], which is probably the most studied ion to date, has a $3d^6$ configuration in its free ion state.

When the Fe (II) ion is coordinated in an octahedral ligand field ($O_h$) to form a metal complex, the degeneracy of the d orbitals is lifted into the $t_{2g}$ triply degenerate and $e_g$ doubly degenerate states. The difference between these two levels depends on the strength of the ligand field, and the six electrons can then be distributed in two different ways: (i) in the case of a strong field, the six electrons pair up in the $t_{2g}$ orbitals, which leads to an LS diamagnetic and colored state for which the spin $S = 0$ and for which the electronic configuration is $t_{2g}^6 e_g^0$; (ii) in the case of a weak field, the electrons are distributed over the d orbitals according to Hund's rule, leading to a HS paramagnetic and colorless state for which the spin $S = 2$ and for which the electronic configuration is $t_{2g}^4 e_g^2$. This second configuration results in an expansion of the Fe-ligand distance with respect to the LS state, and it is therefore accompanied by an increase of the unit cell volume of the material [27].

According to the type (hydrogen bonding, electrostatic, or pi–pi stacking) and intensity of interactions between molecules, spin transition compounds can exhibit different kinds of transitions with a change in temperature: continuous or gradual transformation while respecting the simple Boltzmann-type statistic, sharp first-order with or without hysteresis cycle, two or multi-steps transitions, or incomplete transitions with a quenched residual HS fraction at a low temperature. These five behaviors have been observed in Fe (II) compounds, and they can occur at low, high, or room temperature. First-order phase transitions accompanied by hysteresis are observed in highly cooperative systems in which intermolecular interactions are strong enough. Historically, König et al. [28] were the first to observe a hysteresis loop in 1976 on the thermally-induced spin transition of the [Fe (4, 7-(CH3)$_2$-phen)$_2$ (NCS)$_2$] compound. Later on, a two-step spin transition was revealed in a Fe (III) complex [29]. It is worth mentioning that during the last ten years, SCO Fe(II) complexes have been synthesized that are characterized by spin transition occuring in three steps [30–32].

To try to explain and reproduce these switching phenomena from the LS to HS states, researchers have proposed different models based on Ising-like [33–36] and atom-phonon [37] descriptions, and, more recently mechano-elastic [38] and electroelastic [39] models have been developed.

A crucial issue concerning SCO nanoparticles is the reduction of size. Indeed, when the surface-to-volume ratio increases, it results in an increasingly large effect on the thermal properties of SCO nanoparticles dominated by interactions between the surrounding environment and the molecules located on the edges (or surfaces).

Experimentally, it has been observed that a drastic change in the properties and, particularly, in the cooperativity of the nanoparticle can occur when the size is smaller than a threshold "limit" size. For example, in the context of a thermo-induced transition, this can lead to a change in the transition temperature of several tens of Kelvins.

In the context of Ising-like models and 1D SCO systems embedded in matrixes, a theoretical study based on an Ising-like model [33,34,40] was presented by Chiruta et al. [41] to understand the physical origins of the three-step behavior. Three-state thermal behavior was also found by S.E. Allal et al. [42]. In 2D SCO compounds, S.E. Allal et al. [43,44] studied two-step transitions and analyzed the conditions to obtain three-state behavior.

Recently, we proposed a homogenous local mean-field approach to study both shape and size effects on 3D SCO cubic and cuboid lattices [45]. This study highlighted three steps of switching and three states under the effect of temperature, as well as two transition stages and three states under the effect of pressure.

In the present study, we focused on the modeling of thermal effects in 2D hexagonal-shaped SCO nanoparticles. The Ising-like model was used and solved in the framework of local mean field approximation (LMFA) in an attempt to reveal the role of interactions with the external environment (matrix). In a hexagonal-shaped lattice, the molecules have more interactions than in a square lattice. This means that due to the lattice topology, all interactions are locally stronger for a hexagonal symmetry than a square symmetry. In this study, we show that interactions with the environment are important, even in a

hexagonal-shaped model. As a result, this symmetry allows one to enhance the effects of the interactions and surfaces on the physical properties of the studied nanoparticles.

This paper is organized as follows. Section 2 is devoted to the model and the calculation method, Section 3 presents the results and a discussion, and Section 4 is the conclusion.

## 2. Model

The first microscopic model simulating the spin transition was introduced by Wajnflasz and Pick [33], who only considered short-range interactions between the spin-crossover sites. This model was reconsidered by Bousseksou et al. [34] in the mean-field approach involving two "antiferromagnetically" coupled sublattices to reproduce a double-step spin transition. Later, Linares et al. [40] added long-range interactions to the Ising-like model to trigger the phenomenon of hysteresis in 1D compounds. Chiruta et al. [41] included an energetic contribution (*L*). In SCO nanoparticles, this term allows one to explicitly deal with interactions between the molecules located on the periphery and in close contact with the environment.

As a result of these previous contributions, we considered short-range, long-range, and matrix effects. The Hamiltonian of the global system for SCO compounds can be written as follows:

$$H = \frac{\Delta - k_B T \ln(g)}{2} \sum_{i=1}^{N_T} \sigma_i - J \sum_{\langle i,j \rangle} \sigma_i \sigma_j - G \sum_{i=1}^{N_T} \sigma_i \langle \sigma \rangle - \sum_{i=1}^{M} L \sigma_i \tag{1}$$

The first term of Equation (1) corresponds to the temperature-dependent field. The ligand field $\Delta$ ($> 0$) is the energy gap between the HS and the LS states that depends on the symmetry and nature of the ligands involved in the complex. The total number of molecules is $N_T$. $T$ is the absolute temperature of the system, $k_B$ is the Boltzmann's constant such as $\beta = 1/(k_B T)$, and $g = g_{HS}/g_{LS}$ is the ratio between the degeneracy of the HS state ($g_{HS}$) and the degeneracy of the LS state ($g_{LS}$). A fictitious spin operator $\sigma$ is associated with each molecule; its eigenstates are $+1$ when the molecule is in the HS state and $-1$ when the molecule is in the LS state [11,46,47]. The second term accounts for the strength of short-range interactions, *J*, limited to first-neighbors. The third term, *G*, pertains to the long-range part of the interaction, and the last term, *L*, equivalent to an applied pressure on the molecules at the surface, is a coupling term related to the interactions between the environment and the atoms located at the corners and at the edges that vanishes for bulk molecules. The *J* and *G* parameters are ferromagnetic-type (>0) interactions and thus favor HS–HS and LS–LS pairs. The ligand field of small spin-crossover nanoparticles may continuously vary between the surface and the core of the nanoparticles. Obviously, the ligand field is smaller at the surface due to the immediate environment of SCO sites, which might be composed of water molecules or event impurities (pollution). In contrast, in the bulk, all SCO metal's centers (iron (II)) are surrounded by nitrogen atoms that locally enhance their ligand field. One may expect a variation of the average ligand field of the nanoparticles as a function of size. To simplify, we only considered the change of the ligand field at the surface in this model and assumed that it remained unchanged inside the lattice or that this variation was negligible at the nanometer scale.

In this study, three types of sites were considered: the atoms located in the bulk ($N_b$) that are surrounded by six first-neighbors and those located on the edge ($N_e$) and the corner ($N_c$) that only interact with four and three first-neighbors, respectively. The molecules located on the surface (edge and corner) therefore have specific properties because they also interact with an external environment (matrix effect).

The case of a system comprising 19 molecules (designated H3) is represented in Figure 1 below.

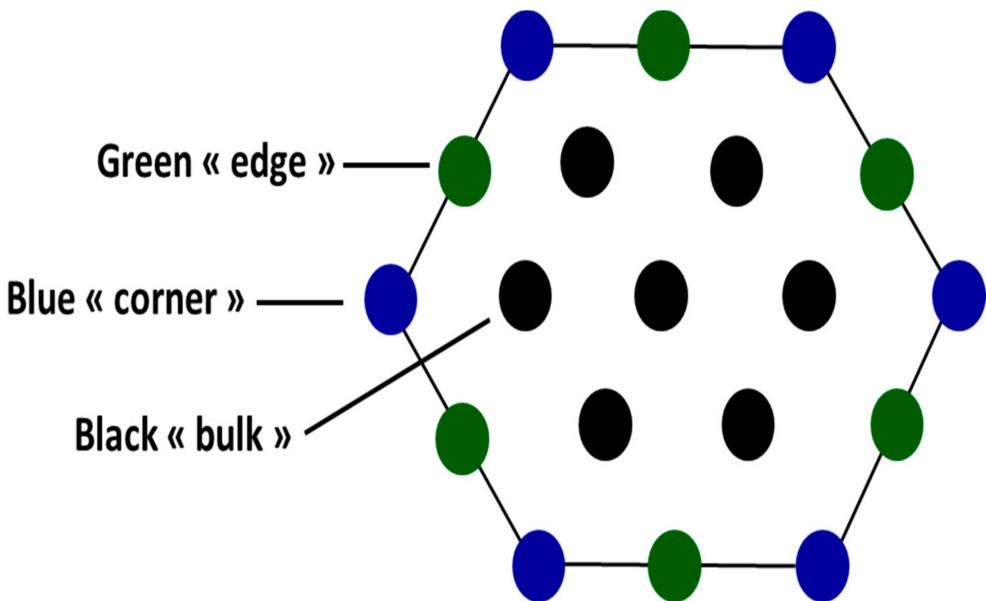

**Figure 1.** Schematic representation of a 2D hexagonal-shaped lattice comprising 19 molecules (H3). Filled black circles represent bulk sites ($N_b$), and filled green and blue circles represent, respectively, the edge ($N_e$) and corner ($N_c$) sites that interact with their immediate environment (matrix).

Considering the three regions in the lattice (corner, edge, and bulk) led us to distinguish three local average order parameters $\langle \sigma_\alpha \rangle$ associated with each of these regions, which brought us to consider a local Hamiltonian for each region $\alpha$ with $\alpha = b, c,$ and $e$ corresponding to the bulk, corner, and edge, respectively, in the local mean field approximation.

$$H_\alpha = \frac{\Delta - k_B T \ln(g)}{2} \sum_{i=1}^{N_\alpha} \sigma_i - J \sum_{i=1}^{N_\alpha} \sigma_i \, q_\alpha \langle \sigma_\alpha \rangle - G \sum_{i=1}^{N_\alpha} \sigma_i \langle \sigma \rangle - \sum_{i=1}^{N_\alpha} L z_\alpha \sigma_i \qquad (2)$$

The number of interactions, $J$, between a molecule and its first-neighbors is denoted as $q_\alpha$, and the number of interactions, $L$, between a molecule and the external environment is referred by $z_\alpha$.

For each region, the Hamiltonian can be cast as Equation (3) below:

$$H_\alpha = \sum_{i=1}^{N_\alpha} \frac{\Delta - k_B T \ln(g) - 2\, J\, q_\alpha \langle \sigma_\alpha \rangle - 2\, G \langle \sigma \rangle - 2\, L\, z_\alpha}{2} \sigma_i = -\sum_{i=1}^{N_\alpha} h_\alpha \sigma_i \qquad (3)$$

with

$$h_\alpha = -\frac{\Delta - k_B T \ln(g) - 2\, J\, q_\alpha \, \langle \sigma_\alpha \rangle - 2\, G \langle \sigma \rangle - 2\, L\, z_\alpha}{2} \qquad (4)$$

The characteristics of the various local situations considered in the model are listed in Table 1.

**Table 1.** The ligand field contributions as a function of various localizations in the hexagonal-shaped lattice.

| Site | Bulk | Edge | Corner |
|------|------|------|--------|
| $q_\alpha$ | 6 | 4 | 3 |
| $z_\alpha$ | 0 | 2 | 3 |
| ligand-field | $\frac{\Delta - k_B T \ln(g)}{2}$ | $\frac{\Delta - k_B T \ln(g) - 4L}{2}$ | $\frac{\Delta - k_B T \ln(g) - 6L}{2}$ |

The partition function of this inhomogeneous mean-field system comprising $N_b$, $N_c$, and $N_e$ atoms belonging to the bulk, corner, and edge regions, respectively, is written as:

$$Z = Z_b Z_c Z_e \tag{5}$$

The partition functions $Z_b$, $Z_c$, and $Z_e$—connected to the bulk, corner, and edge, respectively—have the general form:

$$Z_\alpha = 2\cosh\beta\left(\frac{\Delta - k_B T\ln(g) - 2\,J\,q_\alpha\,\langle\sigma_\alpha\rangle - 2\,G\;\langle\sigma\rangle - 2\,L\,z_\alpha}{2}\right) \text{ with } \alpha = b, c, e \tag{6}$$

The average value of the fictitious spin of the system is calculated as a weighted average of the three order parameters of the three regions: bulk, corner, and edge.

$$\langle\sigma\rangle = \frac{N_b\langle\sigma_b\rangle + N_c\langle\sigma_c\rangle + N_e\langle\sigma_e\rangle}{N_T} = \frac{1}{N_T}\sum_\alpha N_\alpha\,\langle\sigma_\alpha\rangle \tag{7}$$

The high-spin fraction, $Nhs$, which is the probability that the HS state is occupied, is given by:

$$Nhs = \frac{1 + \langle\sigma\rangle}{2} \tag{8}$$

Using the Gibbs–Bogoliubov–Feynman inequality, a variational approximation of the upper bound Helmholtz free energy could be derived, such that the total mean-field free energy per site is written as:

$$F = \sum_\alpha F_\alpha \quad F = \sum_\alpha F_\alpha \text{ with } \alpha = b, c, \tag{9}$$

with

$$F_\alpha = \frac{q_\alpha N_\alpha}{2 N_{tot}}\Gamma < \sigma_\alpha >^2 - k_B T\ln Z_\alpha, \; \alpha = b, c, e \tag{10}$$

Then:

$$\sigma_\alpha = \tanh(\beta\,h_\alpha) = -\tanh\beta\left(\frac{\Delta - k_B T\ln(g) - 2\,J\,q_\alpha\,\langle\sigma_\alpha\rangle - 2\,G\;\langle\sigma\rangle - 2\,L\,z_\alpha}{2}\right) \tag{11}$$

Thus, for each region of b, c, and e, the average spin state is:

$$\langle\sigma_b\rangle = \tanh\left(-\frac{\Delta - k_B\,T\ln(g) - 2\times J\times 6\,\langle\sigma_b\rangle - 2\times 0\times L}{2\,k_B T}\right)$$

$$\langle\sigma_c\rangle = \tanh\left(-\frac{\Delta - k_B\,T\ln(g) - 2\times J\times 3\,\langle\sigma_c\rangle - 2\times 3\times L}{2\,k_B T}\right) \tag{12}$$

$$\langle\sigma_e\rangle = \tanh\left(-\frac{\Delta - k_B\,T\ln(g) - 2\times J\times 4\,\langle\sigma_e\rangle - 2\times 2\times L}{2\,k_B T}\right)$$

which is equivalent to:

$$\langle\sigma_b\rangle - \tanh\left(-\frac{\Delta - k_B\,T\ln(g) - 2\times J\times 6\,\langle\sigma_c\rangle - 2G\sigma - 2\times 0\times L}{2\,k_B T}\right) = f_3(\langle\sigma_c\rangle, \langle\sigma_e\rangle, \langle\sigma_b\rangle) = 0$$

$$\langle\sigma_c\rangle - \tanh\left(-\frac{\Delta - k_B\,T\ln(g) - 2\times J\times 3\,\langle\sigma_c\rangle - 2G\sigma - 2\times 3\times L}{2\,k_B T}\right) = f_1(\langle\sigma_c\rangle, \langle\sigma_e\rangle, \langle\sigma_b\rangle) = 0 \tag{13}$$

$$\langle\sigma_e\rangle - \tanh\left(-\frac{\Delta - k_B\,T\ln(g) - 2\times J\times 4\,\langle\sigma_c\rangle - 2G\sigma - 2\times 2\times L}{2\,k_B T}\right) = f_2(\langle\sigma_c\rangle, \langle\sigma_e\rangle, \langle\sigma_b\rangle) = 0$$

These equations are combined with the relation (Equation (1)):

$$\langle\sigma\rangle = \frac{N_b\langle\sigma_b\rangle + N_c\langle\sigma_c\rangle + N_e\langle\sigma_e\rangle}{N_T} = \frac{1}{N_T}\sum_\alpha N_\alpha\,\langle\sigma_\alpha\rangle$$

The problem therefore consisted of finding the three order parameters—$\langle \sigma_b \rangle$, $\langle \sigma_c \rangle$, and $\langle \sigma_e \rangle$—that were simultaneous solutions of the three functions $f_1$, $f_2$, and $f_3$.

This system was solved by the Newton method, which is appropriate for rapid convergences. Calculations were initiated from three points ($x_1$, $x_2$, and $x_3$) close to the solution and that represented, respectively, $\langle \sigma_b \rangle$, $\langle \sigma_c \rangle$, and $\langle \sigma_e \rangle$. Solving the system amounted to calculating $\epsilon_1$, $\epsilon_2$, and $\epsilon_2$ so that:

$$f_i \left( x_1 + \epsilon_1, \quad x_2 + \epsilon_2, \quad x_3 + \epsilon_3 \right) = 0, \ i = 1,\ 2,\ 3 \tag{14}$$

which allowed us to write the following equation in the vicinity of points $x_1$, $x_2$, and $x_3$:

$$f_i \left( x_1 + \epsilon_1, x_2 + \epsilon_2, x_3 + \epsilon_3 \right) \cong f_i \left( x_1, x_2, x_3 \right) + \epsilon_1 \frac{\partial f_i}{\partial x_1} + \epsilon_2 \frac{\partial f_i}{\partial x_2} + \epsilon_3 \frac{\partial f_i}{\partial x_3} = 0, \ i = 1,\ 2,\ 3 \tag{15}$$

Thus:

$$\epsilon_1 \frac{\partial f_i}{\partial x_1} + \epsilon_2 \frac{\partial f_i}{\partial x_2} + \epsilon_3 \frac{\partial f_i}{\partial x_3} = -f_i \left( x_1, x_2, x_3 \right) \tag{16}$$

The following system of three equations with three unknowns was thus obtained.

$$\epsilon_1 \frac{\partial f_1}{\partial x_1} + \epsilon_2 \frac{\partial f_1}{\partial x_2} + \epsilon_3 \frac{\partial f_1}{\partial x_3} = -f_1 \left( x_1, x_2, x_3 \right)$$

$$\epsilon_1 \frac{\partial f_2}{\partial x_1} + \epsilon_2 \frac{\partial f_2}{\partial x_2} + \epsilon_3 \frac{\partial f_2}{\partial x_3} = -f_2 \left( x_1, x_2, x_3 \right) \tag{17}$$

$$\epsilon_1 \frac{\partial f_3}{\partial x_1} + \epsilon_2 \frac{\partial f_3}{\partial x_2} + \epsilon_3 \frac{\partial f_3}{\partial x_3} = -f_3 \left( x_1, x_2, x_3 \right)$$

and $\epsilon_1$, $\epsilon_2$, and $\epsilon_3$ were calculated as follows:

$$\epsilon_1 = \frac{\begin{vmatrix} -f_1 & \frac{\partial f_1}{\partial x_2} & \frac{\partial f_1}{\partial x_3} \\ -f_2 & \frac{\partial f_2}{\partial x_2} & \frac{\partial f_2}{\partial x_3} \\ -f_3 & \frac{\partial f_3}{\partial x_2} & \frac{\partial f_3}{\partial x_3} \end{vmatrix}}{\begin{vmatrix} \frac{\partial f_1}{\partial x_1} & \frac{\partial f_1}{\partial x_2} & \frac{\partial f_1}{\partial x_3} \\ \frac{\partial f_2}{\partial x_1} & \frac{\partial f_2}{\partial x_2} & \frac{\partial f_2}{\partial x_3} \\ \frac{\partial f_3}{\partial x_1} & \frac{\partial f_3}{\partial x_2} & \frac{\partial f_3}{\partial x_3} \end{vmatrix}} \quad \epsilon_2 = \frac{\begin{vmatrix} \frac{\partial f_1}{\partial x_1} & -f_1 & \frac{\partial f_1}{\partial x_3} \\ \frac{\partial f_2}{\partial x_1} & -f_2 & \frac{\partial f_2}{\partial x_3} \\ \frac{\partial f_3}{\partial x_1} & -f_3 & \frac{\partial f_3}{\partial x_3} \end{vmatrix}}{\begin{vmatrix} \frac{\partial f_1}{\partial x_1} & \frac{\partial f_1}{\partial x_2} & \frac{\partial f_1}{\partial x_3} \\ \frac{\partial f_2}{\partial x_1} & \frac{\partial f_2}{\partial x_2} & \frac{\partial f_2}{\partial x_3} \\ \frac{\partial f_3}{\partial x_1} & \frac{\partial f_3}{\partial x_2} & \frac{\partial f_3}{\partial x_3} \end{vmatrix}}$$

$$\epsilon_3 = \frac{\begin{vmatrix} \frac{\partial f_1}{\partial x_1} & \frac{\partial f_1}{\partial x_2} & -f_1 \\ \frac{\partial f_2}{\partial x_1} & \frac{\partial f_2}{\partial x_2} & -f_2 \\ \frac{\partial f_3}{\partial x_1} & \frac{\partial f_3}{\partial x_2} & -f_3 \end{vmatrix}}{\begin{vmatrix} \frac{\partial f_1}{\partial x_1} & \frac{\partial f_1}{\partial x_2} & \frac{\partial f_1}{\partial x_3} \\ \frac{\partial f_2}{\partial x_1} & \frac{\partial f_2}{\partial x_2} & \frac{\partial f_2}{\partial x_3} \\ \frac{\partial f_3}{\partial x_1} & \frac{\partial f_3}{\partial x_2} & \frac{\partial f_3}{\partial x_3} \end{vmatrix}} \tag{18}$$

In a general way, the $f_i$ functions with $i = 1$ (bulk), 2 (corner), 3 (edge) and their derivatives can be expressed as follows:

$$f_i = x_i - \tanh\left( -\frac{\Delta - T \ \ln(g) - 2*q_i \ J \ x_i - 2 \ G \ \sigma - 2*z_i \ L}{2k_B T} \right) \tag{19}$$

$$\frac{\partial f_i}{\partial x_i} = 1 - \left( 1 - \tanh^2\left( -\frac{\Delta - T \ \ln(g) - 2*q_i \ J x_i - 2 \ G \ \sigma - 2*z_i \ L}{2k_B T} \right) \right) \left( \frac{q_i \ J}{k_B T} + \frac{G \ N_i}{N_T k_B T} \right) \tag{20}$$

$$\frac{\partial f_i}{\partial x_j} = -\left(1 - \tanh^2\left(-\frac{\Delta - T \ln(g) - 2 * q_i J x_i - 2 G \sigma - 2 * z_i L}{2k_B T}\right)\right)\left(\frac{G N_i}{N_T k_B T}\right) \quad (21)$$

## 3. Numerical Results and Discussion

In [48], we showed that the size dependence of the equilibrium temperature $T_{eq}$ of such a system could be described analytically, and, more precisely, we reached the conclusion that the equilibrium temperature was the result of a null total effective ligand-field. As a result, and according to the expressions listed in Table 1, it was deduced that the transition takes place at the equilibrium temperature, $T_{eq}$, thus obeying the following constraint:

$$\frac{\Delta - k_B T_{eq} \ln(g)}{2} \times N_b + \frac{\Delta - k_B T_{eq} \ln(g) - 4L}{2} \times N_e + \frac{\Delta - k_B T_{eq} \ln(g) - 6L}{2} \times N_c = 0 \quad (22)$$

which led us to the following expression of the equilibrium temperature:

$$T_{eq} = \frac{N_b}{N_T} T_{eq}^{bulk} + \frac{N_e}{N_T} T_{eq}^{edge} + \frac{N_c}{N_T} T_{eq}^{corner} \quad (23)$$

Here, $T_{eq}^{bulk}$, $T_{eq}^{edge}$, and $T_{eq}^{corner}$ are the respective equilibrium temperatures of the bulk, edge, and corner whose analytical expressions are:

$$T_{eq}^{bulk} = \frac{\Delta}{k_B \ln(g)} \quad T_{eq}^{edge} = \frac{\Delta - 4L}{k_B \ln(g)} \quad T_{eq}^{corner} = \frac{\Delta - 6L}{k_B \ln(g)} \quad (24)$$

This means that, for an infinite size system ($N_T \simeq N_b$, since $N_b >> N_c$ and $N_b >> N_e$) with $L/k_B$ and $\ln(g)$ being constant, $T_{eq}$ tends towards the equilibrium temperature of the bulk $T_{eq}^b = \frac{\Delta/k_B}{\ln(g)}$. Equation (23) shows that the equilibrium temperature of the system has two limiting values, namely $T_{eq}^{corner} = \frac{\Delta - 6L}{k_B \ln(g)}$ and $T_{eq}^{bulk} = \frac{\Delta}{k_B \ln(g)}$. It also clearly shows that the equilibrium temperature decreases when the number of interactions with the matrix ($L$ term) increases and, therefore, when the size of the lattice decreases. This point is fundamental in understanding and predicting the occurrence of first order transitions with hysteresis loops. Indeed, in a pure Ising model, the order–disorder (or Curie) temperature, $T_{O.D.}$, is obtained from $\Delta/k_B = 0$, $L/k_B = 0$, and $g = 1$ ($\ln(g) = 0$) in the Hamiltonian of Equation (1). First order transitions with hysteresis occur when $T_{O.D.} > T_{eq}$; otherwise, only a gradual transition takes place.

In the following section results of simulations for 2D hexagonal lattices composed of 19, 37, 61, 91 and127 molecules and designated as H3, H4, H5, H6 and H7 respectively are discussed. The ratio parameter $r$ was defined as $r = (N_c + N_e)/N_T$, where $N_T = N_b + N_c + N_e$ is the total number of molecules (Table 2 below).

**Table 2.** Values of the ratio parameter $r$ for different sizes of hexagonal-shaped systems.

| Size of the System | $N_s = N_c + N_e$ | $N_T$ | $r = N_s/N_T$ |
|:---:|:---:|:---:|:---:|
| H3 | 12 | 19 | 0.63 |
| H4 | 18 | 37 | 0.48 |
| H5 | 24 | 61 | 0.39 |
| H6 | 30 | 91 | 0.32 |
| H7 | 36 | 127 | 0.28 |

### 3.1. Case L = 0

First of all, only the short- and long-range interactions of $J$ and G were considered in the calculations, which consisted of setting $L/k_B = 0$ K. The thermodynamic parameters used in the calculations were values typical of the Fe(btr)$_2$(NCS)$_2$ SCO complex [4]: the enthalpy and entropy changes associated with the spin transition were around $\Delta H \approx 11$ kJ/mol and

$\Delta S \approx 50$ J/mol/K, which led to $\ln(g) = \Delta S/R \approx 6.01$ (where R is the perfect gas constant) and $\Delta = \Delta H/R \approx 1300$ K. The resulting equilibrium temperature value of the system was $T_{eq} = \frac{\Delta}{(k_B \ln(g))} \approx 216$ K (in Figure 2, $T_{eq} = 216.3$ K), and according to Equation (24), it also corresponded to the bulk transition temperature. At this temperature, there were as many molecules in the LS state as there are molecules in the HS state, and the HS molar fraction was $Nhs = \frac{1}{2}$.

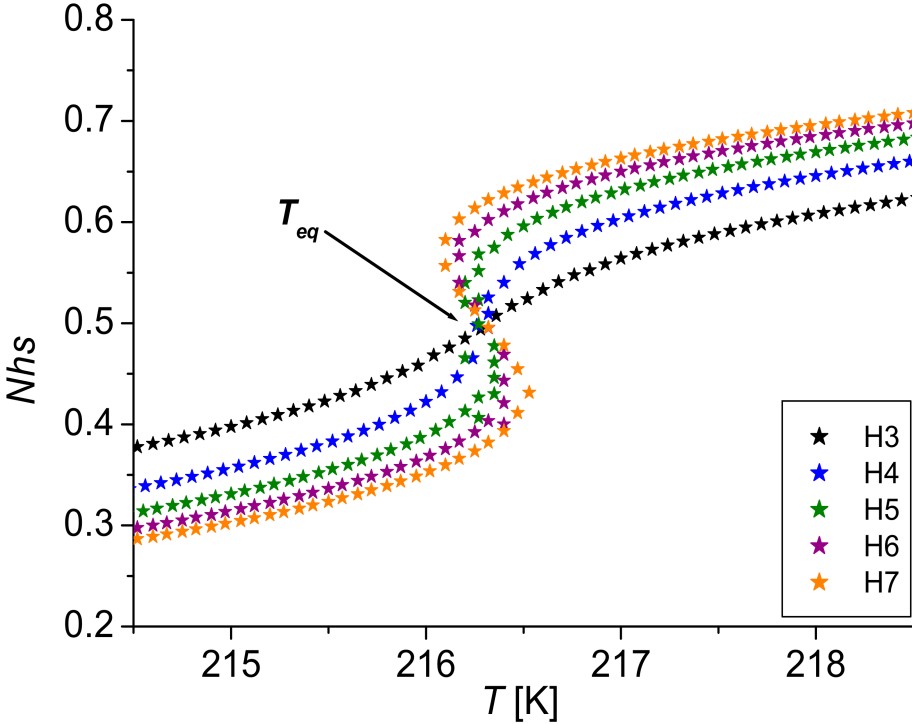

**Figure 2.** Thermal evolution of the HS molar fraction for different lattice sizes in the 2D hexagonal lattice. Black stars: H3 (19 atoms); blue stars: H4 (37 atoms); green stars: H5 (61 atoms); purple stars: H6 (91 atoms); and orange stars: H7 (127 atoms). The computational parameters were: $\Delta/k_B = 1300$ K, $J/k_B = 30.20$ K, $G/k_B = 49.70$ K, $L/k_B = 0$ K, and $\ln(g) = 6.01$.

The calculations were carried out for systems H3–H7 (19–127 atoms) and are reported in Figure 2.

As can be seen in Figure 2, the equilibrium temperature $T_{eq}$ was independent of the system's size. This result was directly linked to the absence of interactions with the surrounding matrix. On the other hand, different behaviors were observed with regard to the type of spin transition. Though the transition was gradual for the H3 and H4 lattices, a thermal hysteresis loop appeared for the H5, H6, and H7 systems.

Moreover, the width of the thermal hysteresis $\Delta T = T_{up} - T_{down}$ increased as the ratio $r = N_s/N_T$ decreased. In Figure 3, the plot of T against the size $H_i$ clearly reveals a linear change of the thermal hysteresis width as a function of size.

These results are explained by the fact that by increasing the size, the coupling between the molecules was increased and the order–disorder transition temperature $T_{O.D.}$ of the hexagonal system (or Curie temperature) therefore increased. For H5, H6, and H7, $T_{eq} < T_{O.D.}$, which led to a first-order phase transition. By increasing the ratio $r$ (See Table 2), i.e., by increasing the number of molecules on the surface, $T_{O.D.}$ decreased. The condition $T_{O.D.} < T_{eq}$ was also satisfied, and a gradual transition took place for H3 and H4. The hysteretic behavior was therefore strongly impacted by the decrease in the number of molecules of the SCO compound.

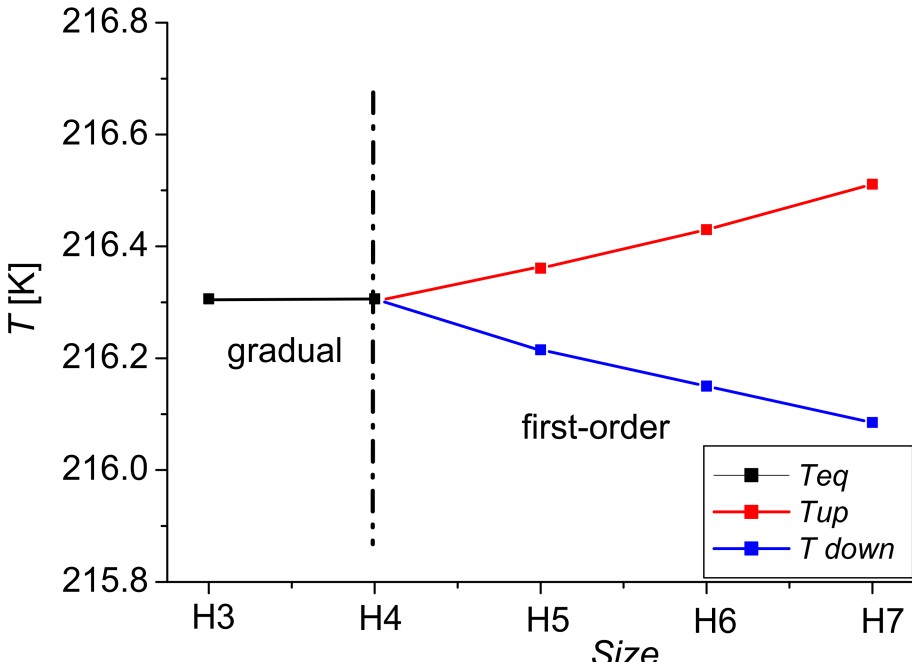

**Figure 3.** Lower ($T_{down} = T_{HS} - T_{LS}$) and upper ($T_{up} = T_{LS} - T_{HS}$) transition temperatures of the total HS fraction of Figure 2 as a function of the size of the system.

### 3.2. Case L ≠ 0

After considering the strength of interactions between the boundary molecules and the matrix, the 2D SCO went from gradual to three-step behaviors. It is interesting to remark that a two-step behavior with three states was also obtained for the same temperature

#### 3.2.1. Three Steps

The value of the $L/k_B$ parameter was gradually increased from 0 to 160 K, and calculations were made for the H6 (91 atoms) system It is worth mentioning that modifying the values of L/kB correspond to a change in the interaction strength between the nanoparticle and its environment (matrix effect) which is achieved with a change in the chemical nature of the matrix. The results are reported in Figure 4.

Table 3 provides the values of the transition temperatures of the different regions (corner, edge, and bulk) and the overall system for different coupling strengths with the matrix, as derived from the numerical simulations.

**Table 3.** Equilibrium temperatures of the corner, edge, and bulk calculated for different values of the *L* parameter for the H6 system (91 atoms). The computational parameters were $\Delta/k_B$ = 1300 K, $J/k_B$ = 40 K, $G/k_B$ = 70 K, and $\ln(g)$ = 6.01.

| $L/k_B$[K] | $T_{eq}^{corner}$[K] | $T_{eq}^{edge}$[K] | $T_{eq}^{bulk}$[K] | $T_{eq}(LMFA)$[K] |
|---|---|---|---|---|
| 0 | 216.30 | 216.30 | 216.30 | 216.30 |
| 50 | 166.38 | 183.02 | 216.30 | 204.23 |
| 120 | 96.50 | 136.43 | 216.30 | 187.33 |
| 160 | 56.57 | 109.81 | 216.30 | 177.68 |

As shown in Figure 4, the value $L/k_B$ = 0 K led to a single thermal hysteresis loop at the equilibrium temperature $T_{eq} = T_{eq}^{bulk} \approx$ 216.3 K. For a slightly higher value of $L/k_B$ = 50 K, boundary effects increased and a two-step transition emerged. A complete three-step transition was achieved for strong coupling with the environment, such as for $L/k_B$ = 160 K. Figure 5 below highlights the three steps associated with three hysteresis loops HYST1, HYST2, and HYST3 obtained in the heating and cooling modes and that

were distinct in the sense of the absence of overlapping hysteresis loops. The stable states were: (i) for T < T', the states with *Nhs* close to 0; (ii) for temperature values between *T'* and *T''*, the states with *Nhs* close to 0.06; (iii) for the region between *T''* and *T'''*, the states with *Nhs* close to 0.31; and (iv) for T > *T'''*, the states with *Nhs* > 0.8.

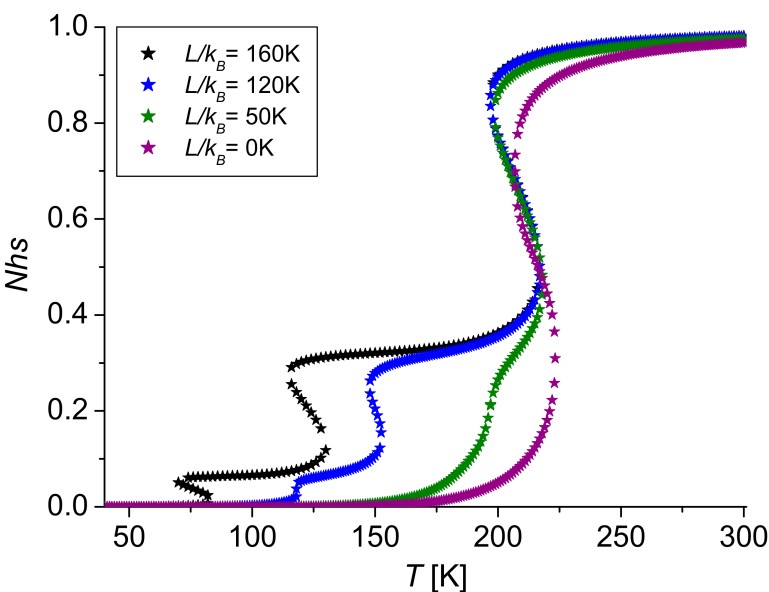

**Figure 4.** The simulated thermal evolution of the HS fraction *Nhs* for different values of the $L/k_B$ parameter for the H6 system (91 atoms). Black stars: $L/k_B$ = 160 K; blue stars: $L/k_B$ = 120 K; green stars: $L/k_B$ = 50 K; and purple stars: $L/k_B$ = 0 K. The computational parameters were: $\Delta/k_B$ = 1300 K, $J/k_B$ = 40 K, $G/k_B$ = 70 K, and ln($g$) = 6.01.

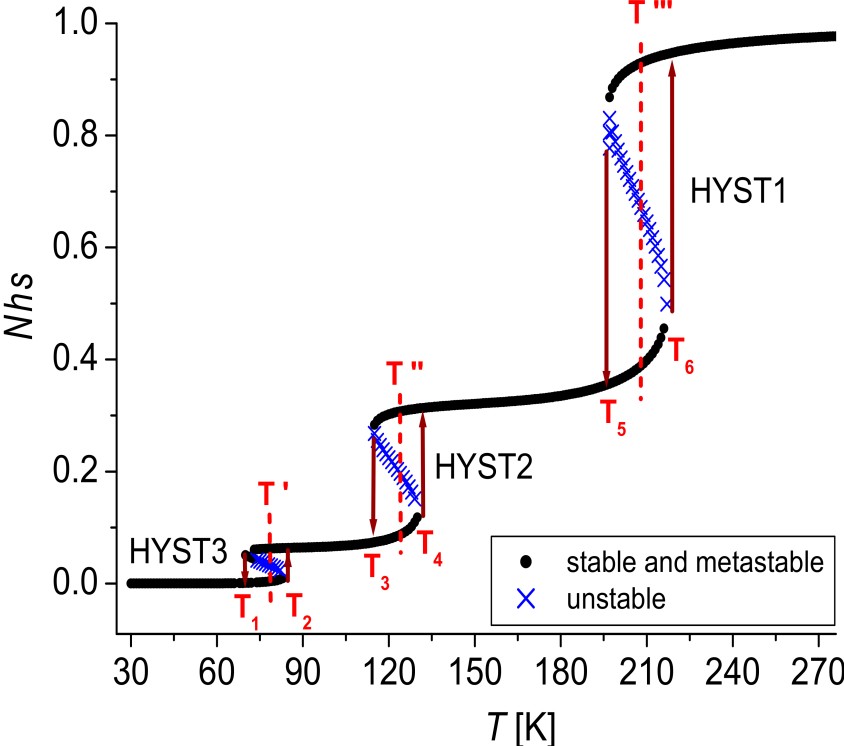

**Figure 5.** Thermal evolution of the HS molar fraction showing three hysteresis loops—HYST1, HYST2, and HYST3—for the H6 case (91 atoms). Black filled circles represent stable and metastable regions, whereas blue crosses correspond to unstable regions. The computational parameters were $\Delta/k_B$ = 1300 K, $J/k_B$ = 40 K, $G/k_B$ = 70 K, $L/k_B$ = 160 K, and ln($g$) = 6.01.

We note that, on the one hand, increasing the value of the $L/k_B$ parameter increased the width of the HYST3 and HYST2 hysteresis loops; on the other hand, a concomitant shift towards lower temperatures led to the presence of two well-defined intermediate plateaus.

The first hysteresis loop, HYST3, was in the temperature range of 67–80 K and, as can be seen in Table 3, was close to the transition temperature of the corner, $T_{eq}^c = \frac{\Delta - 6L}{k_B \ln(g)} \approx$ 57 K, with a plateau at $Nhs \approx 0.06$. This means that at this temperature, a very small number of molecules switched to the HS state. The second hysteresis, HYST2, in the temperature range of 114–130 K, was close to the temperature transition of the edge, $T_{eq}^e = \frac{\Delta - 4L}{k_B \ln(g)} \approx 110$ K, with a plateau at $Nhs \approx 0.31$. As for HYST1, between 197 and 218 K, it corresponded to the bulk transition temperature, $T_{eq}^b = \frac{\Delta}{k_B \ln(g)} = 216.3$ K.

### 3.2.2. Three States

Figure 6 exhibits two hysteresis loops. The first one, denoted HYST1, was in the temperature range of 132–236 K, with $Nhs = 0.32$–1.0, and the second one, HYST2, was in the temperature range of 205–217 K, with $Nhs = 0.15$–0.32.

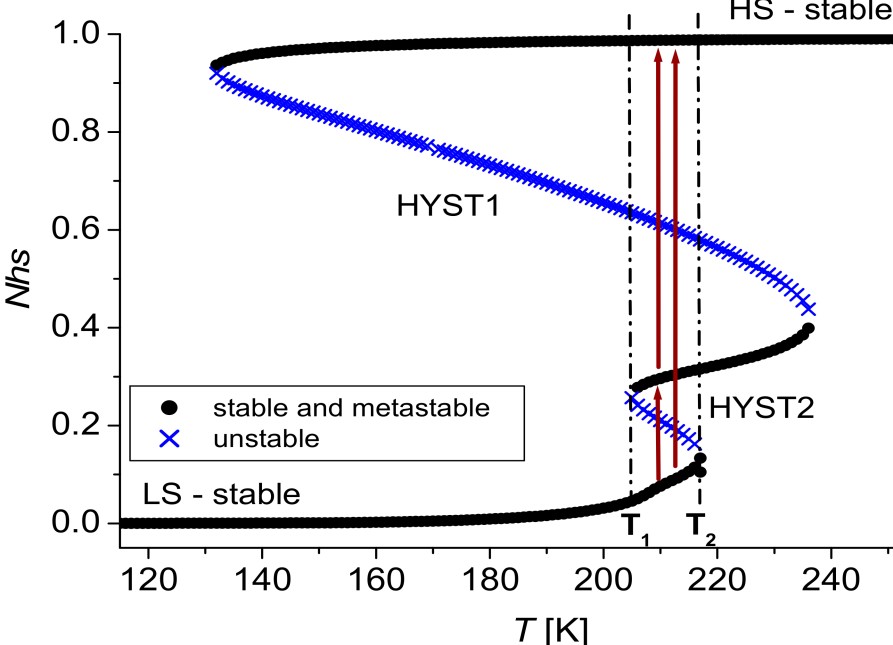

**Figure 6.** Thermal evolution of the HS molar fraction showing three-state behavior in a 2D hexagonal-shaped system comprising 91 atoms (H6). The computational parameters were $\Delta/k_B = 1300$ K, $J/k_B = 57$ K, $G/k_B = 150$ K, $L/k_B = 52$ K, and $\ln(g) = 6.01$.

The two hysteresis loops overlapped over a temperature interval of $T_2 - T_1 \approx 12$ K, which led to the presence of three stable states for the same temperature. These states could be achieved through an appropriate variation of temperature or some other physical stimuli such as light. The intermediate plateau obtained between 205 and 236 K with $Nhs$ between 0.27 and 0.40 was a mixture of the LS and HS configurations, with the majority of molecules in the low-spin state. The branches (blue crosses in Figure 6) between $Nhs = 0.15$–0.26 and $Nhs = 0.40$–0.91 are the unstable states that were not experimentally observable.

Note that these results were obtained by explicitly considering the edge and corner effects at the surface. These results were different from those presented in [11,48], for which the latter effects were not explicitly considered and which led to an absence of the overlapping of hysteresis loops.

## 4. Phase Diagram

The previous results are summarized in Figure 7 by plotting the phase diagram of the system in the space coordinates, temperature versus $L/k_B$. The transition lines corresponding to the limiting branches of the thermal hysteresis when the transitions were of first-order or to the equilibrium temperature when the transitions were continuous (gradual) are represented. The present phase diagram was made for the lattice size H6 (Figures 4 and 5) and has the advantage of clearly highlighting the existence of four regions of thermal behavior as a function of the surface parameter $L$. Thus, in region (i) corresponding to $L/k_B < 30$ K, the system presented a single first-order phase transition (equilibrium transition temperature: $T_{eq1} \sim \frac{T_{1up}+T_{1down}}{2}$), while in region (ii), obtained for $30 < L/k_B < 90$, the transition from LS to HS involved a gradual transition (equilibrium transition temperature: $T_{eq2}$) followed by a hysteretic first-order transition of region (i). Above $L/k_B = 90$ K and below $L/k_B = 130$ K, i.e., region (iii), the previous continuous transition became first-order with limiting transition temperatures, $T_{2up}$ and $T_{2down}$, and a new bifurcation appeared, leading to the emergence of a third transition line of a gradual phase transition with equilibrium temperature $T_{eq3}$. In the fourth region (iv), i.e., for $L/k_B > 130$ K, all previous phase transitions became of the first order, and then the system exhibited three consecutive first-order transitions. Remarkably, the phase diagram of Figure 7 shows that as the $L/k_B$ value decreased, the widths of the hysteresis loops $(T_{3up} - T_{3down})$ and $(T_{2up} - T_{2down})$—respectively associated with HYST3 and HYST2 and corresponding to the corner and edge contributions—linearly decreased with $L/k_B$, while that of the bulk $(T_{1up} - T_{1down})$ remained unchanged except for very weak $L/k_B$ values. Overall, the different regions of this phase diagram confirm that nano-switchable systems combining competing interactions between surface environment and bulk effects may lead to rich thermodynamic behaviors, going from a single first-order transition (region (i)) to consecutive multi-step first-order phase transitions (region (iv)).

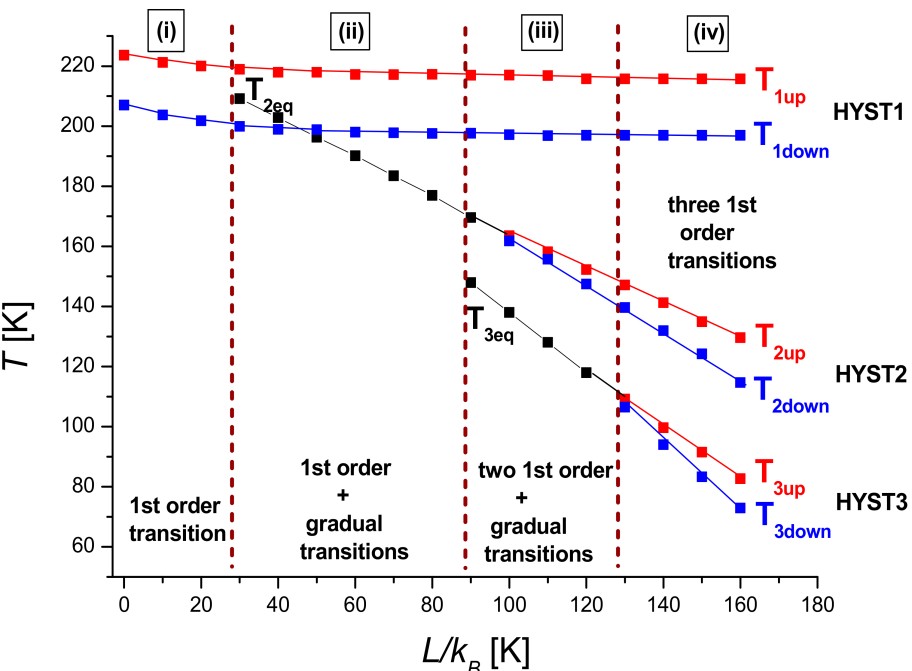

**Figure 7.** Phase diagram $T = f(L/k_B)$ for a 2D hexagonal-shaped system comprising 91 atoms (H6). The red and blue squares correspond, respectively, to the upper and lower transitions for the heating ($T_{up}$) and cooling ($T_{down}$) temperatures of the thermal HS fraction. The black squares correspond to the equilibrium temperatures ($T_{eq}$) of the gradual transitions. The computational parameters were $\Delta/k_B = 1300$ K, $J/k_B = 40$ K, $G/k_B = 70$ K, and $\ln(g) = 6.01$.

## 5. Conclusions

Local mean field approximation was applied to study the properties of SCO nanostructures using the Ising-like model. The results showed the possibility of three types of transition for the five 2D hexagonal systems that were studied: continuous, gradual, or with two or three hysteresis cycles in three or four steps, with overlapping cycles in the case of three steps. These different properties, as functions of temperature, were interpreted in terms of the competition between internal interaction coupling, as modeled by the J and G parameters, and external interaction coupling, as modeled by L. The effects of these terms were dependent on both the size and ratio $r$, and the overall effect was a different shift of the order–disorder transition temperature $T_{O.D.}$ and the equilibrium temperature $T_{eq}$. These results are to be compared to the re-entrance effect that impacts the type of transition that occurs as a result of this competition. Finally, a phase diagram in the variables of T v/s $L/k_B$ clearly highlights the existence of four regions of thermal behavior as a function of the surface parameter $L/k_B$.

**Author Contributions:** Coneptualization, J.L. and K.B.; methodology, J.L., K.B., M.N.; software C.C., J.L., M.N.; validations, J.L., K.B., P.-R.D.; formal analysis, J.L., K.B., C.C., M.N., P.-R.D.; writing-original draft preparation, C.C., P.-R.D., K.B., J.L.; writing-review and editing: K.B., J.L., C.C., P.-R.D., M.N. All authors have read and agreed to the published version of the manuscript.

**Funding:** This research received no external funding.

**Data Availability Statement:** Not applicable.

**Acknowledgments:** This work was supported by the French-Japan LIA (International Associate laboratory), the ANR project Mol-CoSM N° ANR-20-CE07-0028-02, the Universities of Versailles and Paris-Saclay-UPSAY, and the CNRS (Centre National de la Recherche Scientifique) and Ambassade de France au Sénégal. We thank all of them for their financial support.

**Conflicts of Interest:** The authors declare no conflict of interest.

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
