# Peer review of "Hexagonal-Shaped Spin Crossover Nanoparticles Studied by Ising-Like Model Solved by Local Mean Field Approximation"

_magnetochemistry, doi:10.3390/magnetochemistry7050069_

Round 1
Reviewer 1 Report
The authors have theoretically studied the magnetic properties of hexagonal-shaped spin crossover (SCO) nanoparticles by using the Ising-like model with local mean field approximation. Their results are interpreted by the competition between the order/disorder temperature TO.D. and the equilibrium temperature Teq from the ligand field. Their model is carefully introduced, and their results and conclusions seem plausible. I would recommend this article to be published in ‘magnetochemistry’ if the authors could answer to my questions/comments below.
(i)2D hexagonal lattice
The reason why the authors focused on the hexagonal shaped model was not clear to me (p.2 line 97-100). What is the biggest difference comparing to the model with a simple square lattice? More interaction? More edge/corner effect? or Do the actual materials, such as the [Fe(4,7-(CH3)2-phen)2(NCS)2], take a hexagonal structure? It would be nice if the authors could give some insight in the introduction.
(ii) Experimental evidence
In my understanding, the SCO of the bulk material occurs due to the molecular deformation (change of the ligand field) at the spin site with external stimulus. However, the ligand field Δ is fixed in the authors’ calculation, and the transition of the spin state seem to be decided by the competition of transition temperature from several spin sites (bulk, edge and corner).
Then, my question would be, ‘Is there some experimental evidence that the ligand field do not change for the SCO nanoparticle?’. I would guess the answer would be around the line 79-86 of the manuscript, but there is no reference around those sentences. I would like to ask the authors to give some comments and add several references.
(iii) Thermal hysteresis
I did not understand well the interpretation of the thermal hysteresis ΔT=Tup-Tdown in Fig.2 and Fig.3 (line 243-250). Why the hysteresis becomes larger if the system size increases? Would the hysteresis ΔT=Tup-Tdown go infinite if you have an infinite system?
Moreover, what is the difference in the definitions of stable/metastable and unstable in Figs. 5 and 6? Are there also unstable regions for Fig. 4?
(iv) typos?
Table 2: ‘r=Ns/Ntot’ -> ‘r=Ns/NT’
Table 3 caption: lng -> ln(g)
Reviewer 2 Report
The authors present the results of the modelling of a 2D system of spin crossover molecules. The phenomenological approach is based on Ising-like model, allowing for the (competing) ligand-field and elastic intermolecular interactions, the latter being dependent on the localisation of the SCO centre within a hexagonal 2D arrangment. Considering the the different patterns of the centre-matrix interactions and the different size of the molecular assembly (thus changing the ration of bulk, edge and corner centres) the authors convicingly predict the change from gradual to hysteretic transition as well as appearing of multistep transitions.
The paper deals generally with a quite important subject of the dependence of SCO properties on the size of nano/microcrystals that is vital for the miniaturisation of optical switches used in the future molecular electronic devices.
The paper is more or less written in a way that allow the non-theorist to follow the main reasoning and the message-to-take-home.
I recommend the publication of it in the current form after some small editing corrections (the use of different font sizes in some sentence is not helpful for a reader)
